organic chemistry/materials science

azobenzene, heterocycle, liquid crystal

**Author for correspondence:**
Haiying Zhao
e-mail: hyzhao@imu.edu.cn

This article has been edited by the Royal Society of Chemistry, including the commissioning, peer review process and editorial aspects up to the point of acceptance.

# New azobenzene liquid crystal with dihydropyrazole heterocycle and photoisomerization studies

Xiaoxuan Wang, Zhaoxia Li, Haiying Zhao and Shufeng Chen

Inner Mongolia Key Laboratory of Fine Organic Synthesis, College of Chemistry and Chemical Engineering, Inner Mongolia University, Hohhot 010021, People's Republic of China

HZ, 0000-0003-2891-8684

New azobenzene derivatives with dihydropyrazole heterocycle have been prepared and characterized. According to thermal polarizing microscopy and differential scanning calorimetry studies, the compounds consisting of four linearly linked rings and a long alkoxy chain on the azobenzene side (**3a-8** and **3a-14**) displayed no liquid crystal properties. When the length of mesogenic unit increased to five rings, except for compound **5a-8**, all compounds from **5a-10** to **5a-16** containing a long chain of 10–16 carbon atoms on the side of ester group displayed liquid crystalline properties, and the mesogenic domain gradually narrowed with increase of the chain length. However, in the case of the molecule with long alkoxy chains on both sides, only **5c-16** with a long chain of 16 carbon atoms exhibited liquid crystal behaviour. In addition, these azo compounds underwent isomerization from E to Z under ultraviolet irradiation and then thermal back relaxation slowly in the dark, which can be recycled many times.

## 1. Introduction

Liquid crystal state is an intermediate phase between crystal and isotropic liquid, and includes lyotropic liquid crystal and thermotropic liquid crystal. Thermotropic liquid crystal material, as a special soft material, consists of disc-like, bent-core and rod-like structures. Rod-like thermotropic liquid crystal molecules are composed of linear linked polycyclic (alicyclic or aromatic ring) core and flexible chains. Ester group, imino group, azo group, olefin bond and alkynyl group are usually selected as linkages for aromatic rings [1]. Azobenzene, as one of the traditional photochromic entities, can undergo reversible photoisomerization under ultraviolet and visible light with high thermal stability

**Scheme 1.** Synthesis of compounds **3** and **5**.

[2–4], so azo group is superior to other linkages like Schiffe's base, tolane and ester [5–7]. In recent years, liquid crystal materials of small molecules [8,9] and polymers [10,11] containing azobenzene photosensitive groups have been widely studied for their interesting optical properties. These materials can be used in optical switches [12,13], display technology and optical storage devices [14], etc.

On the other hand, the introduction of heteroatoms with high polarization, such as S, O and N can affect the polarity and geometry of molecules, thus affecting the phase transition temperature, dipole moment, dielectric constant and even the type of liquid crystal phase [15]. Therefore, the liquid crystal with heterocyclic ring plays an important part in the design and synthesis of new functional materials and has become one of the research hotspots in recent years [16–22]. Among them, pyrazole heterocyclic ring has been widely used in material chemistry [23], biological and pharmaceutical industry [24–27] and others because of its high stability and strong dipole moment. Moreover, liquid crystal molecules based on pyrazole with wider mesophase range have also been designed and synthesized [28–34]. If pyrazoles are introduced into azobenzene liquid crystal with excellent optical properties, they may endow some special properties to liquid crystal. However, it has not been reported that azobenzene and pyrazole are integrated into one liquid crystal molecule. In order to obtain novel liquid crystal with good properties, a series of compounds including azobenzene, dihydropyrazole heterocycles and ester linkage were designed (scheme 1), and their optical properties and liquid crystal properties were explored.

# 2. Results and discussion

## 2.1. Thermal behaviour

The liquid crystalline properties of synthesized azobenzene derivatives were investigated by differential scanning calorimetry (DSC) and thermal polarizing microscopy (POM) experiments. The DSC curves are shown in figure 1 and electronic supplementary material, figure S1–S6, and thermal analyses data are listed in table 1. First, two compounds **3a-8** and **3a-14** were studied. Both compounds consisted of four linearly linked rings, in which the methoxy group was on the side near dihydropyrazole and the long chain was on the other side. Unfortunately, neither of the compounds has liquid crystalline properties. From table 1 and electronic supplementary material, figure S1, it can be seen that compound **3a-8** only underwent phase transition from crystal to isotropic liquid during the first heating process, with a

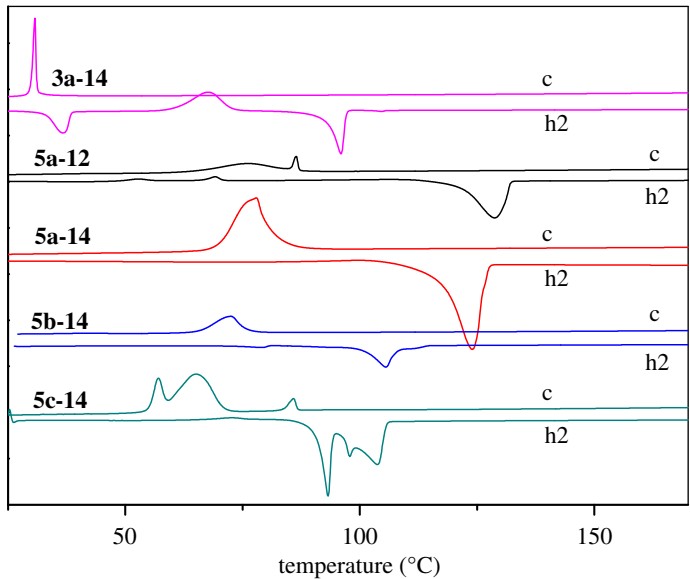

**Figure 1.** DSC curves of selected compounds. c: the first cooling, h2: the second heating.

**Table 1.** Phase transition temperatures and associated enthalpies of compounds **3** and **5**.

| compd. | phase transitions[a] °C ($\Delta H$/kJ mol$^{-1}$) | | | $\Delta T^b$ |
|---|---|---|---|---|
| | first heating | second heating | first cooling | |
| **3a-8** | C 105.6 (49.2) I | $C_1$ 85.7 (-3.2) $C_2$ 100.1 (3.8) I | — | — |
| **3a-14** | $C_1$ 93.1 (16.9) $C_2$ 106.0 (44.6) I | $C_1$ 36.8 (18.4) $C_2$ 67.6 (−32.9)$C_3$ 96.0 (33.9) I | I 30.7 (−18.2) C | — |
| **5a-8** | $C_1$ 141.3 (22.9) $C_2$ 148.8 (19.3) I | $C_1$ 76.0 (−23.5) $C_2$ 128.9 (17.3) $C_3$ 140.0 (12.1) I | I 95.5 (−3.2) $C_2$ 58.4 (−0.3) $C_1$ | — |
| **5a-10** | $C_1$ 125.6 (12.3) $C_2$ 165.8 (48.5) I | $C_1$ 77.7 (−15.8) $C_2$ 119.1 (11.7) $C_3$ 157.7 (26.1) I | I 116.8$^c$ M 95.4 (−1.8) C | 21.4 |
| **5a-12** | C 129.5 (63.2) I | $C_1$ 52.8 (−2.2) $C_2$ 69.2 (−2.2) $C_3$ 128.7 (63.3) I | I 86.5 (−4.9) M 75.9 (−27.2) C | 10.6 |
| **5a-14** | C 126.7 (75.7) I | $C_1$ 123.9 (67.3) I | I 86.0$^c$ M 77.9 (−48.4) C | 8.1 |
| **5a-16** | $C_1$ 77.2 (10.6) $C_2$ 125.5 (23.2) $C_3$ 129.18 (22.9) I | $C_1$ 48. 2 (2.1) $C_2$ 74.1 (−28.9) $C_3$ 125.7 (41.9) I | I 79.8 (−12.5) M 74.2$^c$ $C_2$ 46.8 (−3.3) $C_1$ | 5.6 |
| **5b-10** | $C_1$ 76.6 (0.8) $C_2$ 132.2 (37.1) I | $C_1$ 76.2 (0.7) $C_2$ 132.5 (34.2) I | I 91.2 (−32.7) C | — |
| **5b-14** | $C_1$ 92.4 (6.9) $C_2$ 106.8 (61.7) I | $C_1$ 78.9 (2.2) $C_2$ 105.5 (48.8) I | I 72.4 (−33.3) C | — |
| **5c-10** | $C_1$ 78.8 (9.5) $C_2$ 92.2 (19.4) I | $C_1$ 69.4 (4.5) $C_2$ 75.7 (3.1) $C_3$ 89.7 (21.8) I | I 66.1 (−15.1) $C_2$ 58.6 (−6.7) $C_1$ | — |
| **5c-14** | C 91.9 (64.7) M 104.1 (1.9) I | $C_1$ 93.2 (27.6) $C_2$ 97.8 (13.3) $C_3$ 103.7 (28.5) I | I 85.8 (−3.7) M 65.1 (−40.6) $C_2$ 57.1 (−12.6) $C_1$ | 20.7 |

[a]C: crystal, M: mesophase, I: isotropic liquid.
[b]Mesophase range.
[c]Observed by POM.

melting point of 105.6°C, while no peak was observed in the DSC curve during a slow cooling process. The compound **3a-8** underwent crystallization before melting at the second heating. This phenomenon indicated that the compounds did not crystallize during cooling from isotropic liquids, but in the

further heating process, the molecules were rearranged orderly with the increase of molecular fluidity, and new crystals were formed [35]. After further heating, the compounds melted into isotropic liquids. When the chain length is increased from 8 to 14 carbon atoms, as shown in figure 1 and table 1, the melting point of compound **3a-14** has hardly changed, and the melting–crystallization–melting polycrystalline phase transition has also taken place during the heating process.

Since the above two compounds with different chain lengths have no liquid crystallinity, the number of molecular central rings was increased in the next **5a** series of compounds, and their liquid crystallinity was studied by changing the length of alkyl chain. Compound **5a-8** containing a long chain of eight carbon atoms has no liquid crystallinity. As shown in table 1 and electronic supplementary material, figure S2, two very close endothermic peaks appeared at 141 and 148°C during the first heating process, and the bases of the peaks were superimposed, which was attributed to the process of unstable polycrystalline phase transition from one crystal to another, and then the compound melted. During the cooling process, only relatively low peaks with small enthalpy change occurred near 95 and 58°C, and no mesomorphic phase was observed in POM. During the second heating process, a relatively large exothermic peak of crystallization appeared at 76°C, while two very close endothermic peaks occurred at 129 and 140°C indicating the melting process of compound.

Continuing to increase the length of alkyl chain, we were surprised to find that all compounds from **5a-10** to **5a-16** displayed liquid crystalline behaviours. When the chain length was increased to 10 carbon atoms, the phase transition from one crystal to another occurred in compound **5a-10** during the first heating process (see electronic supplementary material, figure S3). However, compared with **5a-8**, the phase transition temperature range of **5a-10** was widened (125→166°C) and the melting point was increased by 17°C (electronic supplementary material, figure S3). During the cooling process of **5a-10**, the mesomorphic phase was observed from POM image when the temperature dropped to 116.8°C, which lasted to 95.4°C with the mesomorphic range of 21.4°C and then entered the crystalline phase, but the transition peak from isotropic liquid to mesomorphic phase was not observed in the DSC curve perhaps due to the lower enthalpy, and the same phenomenon occurs in the DSC curves of compound **5a-14** and **5a-16**. The POM images of **5a-10** on cooling are shown in figure 2*a* and *b*, which indicated that the molecular arrangement was obviously different in mesomorphic phase (95°C) and crystalline phase (50°C)

The alkyl chain continued to be lengthened to 12 and 14 carbon atoms. Compared with **5a-10**, the melting point of **5a-12** and **5a-14** decreased gradually and tended to form relatively stable crystals. It was shown that there was only a single-phase transition peak during the heating process. However, both compounds exhibited liquid crystalline properties during cooling process. As shown in figure 1 and table 1, compound **5a-12** underwent a small enthalpy change when the temperature dropped to 86.5°C, indicating a transition from isotropic liquid to liquid crystal phase [36]. The liquid crystal phase lasted up to 75.9°C, and the mesomorphic range was 10°C, which was consistent with the result of POM experiment. The POM images of **5a-12** at 83.5°C in mesomorphic phase and 66°C in crystalline phase on cooling are presented in figure 2*c* and *d*. Similar phenomena occurred in compound **5a-14**. It was observed from POM images that **5a-14** began to enter the liquid crystal phase at 86°C and lasted until 80°C. The mesomorphic range of **5a-14** became narrower than that of **5a-12**. The comparison images of **5a-14** at 82.7°C in mesomorphic phase and 42.5°C in crystalline phase on cooling are presented in figure 2*e* and *f*. When the length of alkyl chain was further increased to 16 carbon atoms, the melting point tended to increase, and the unstable polycrystalline phase transition occurred in **5a-16** molecule during the heating process (electronic supplementary material, figure S4). The mesomorphic phase was still observed from POM image of **5a-16** during the cooling process. From table 1, it can be seen that from **5a-10** to **5a-16**, the range of mesomorphic phase gradually narrowed and the clearing point lowered with increase of the chain length. Therefore, lengthening the terminal alkyl chain near the ester group can induce liquid crystallinity of the molecules.

Subsequently, we studied the effect of the chain length near dihydropyrazole on the liquid crystallinity. First, we changed the methoxy group in **5a** series compounds into butoxy group and studied the thermal behaviours. Compared with **5a-10** and **5a-14**, the melting points of **5b-10** (electronic supplementary material figure S5) and **5b-14** (figure 1) were reduced by 30°C and 20°C, respectively, and only a single endothermic or exothermic peak appeared during melting and solidification. However, compounds **5b-10** and **5b-14** had no liquid crystalline properties. Therefore, compounds **5c-10** and **5c-14** with a terminal alkoxy chain containing eight carbon atoms on the side of the dihydropyrazole were prepared. As shown in table 1 and electronic supplementary material, figure S6, the two compounds still exhibited polycrystalline phase transition during heating and cooling, and **5c-10** had no liquid crystal property. Surprisingly, as shown in figure 1 and table 1,

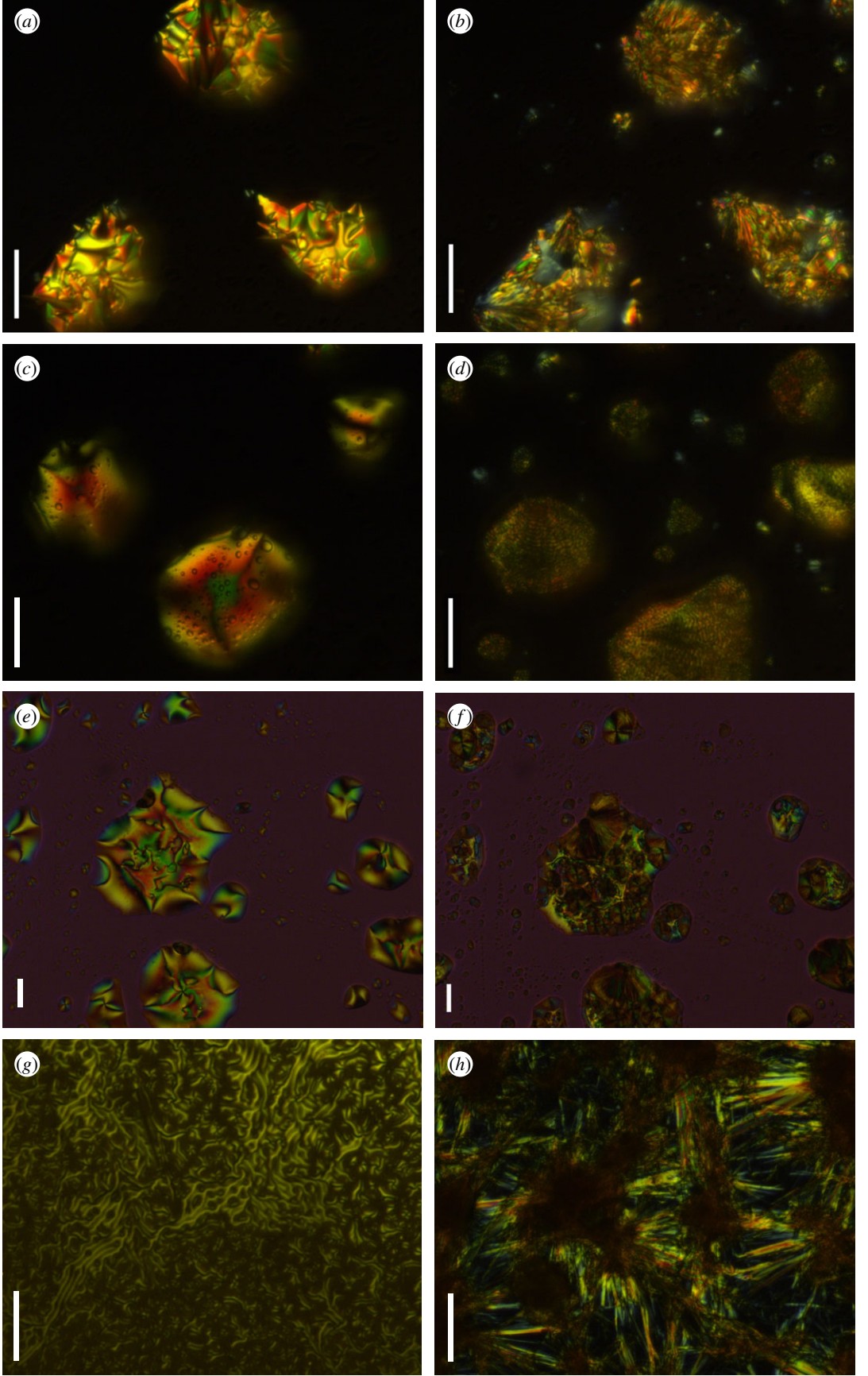

**Figure 2.** POM images **5a-10** (50 ×) at 95℃ (*a*) and 50℃ (*b*), **5a-12** (50 ×) at 83.5℃ (*c*) and 66℃ (*d*), **5a-14** (20 ×) at 82.7℃ (*e*) and 42.5℃ (*f*), **5c-14** (50 ×) at 81.0℃ (*g*) and 44.7℃ (*h*) on cooling. Scale bar, 30 μm.

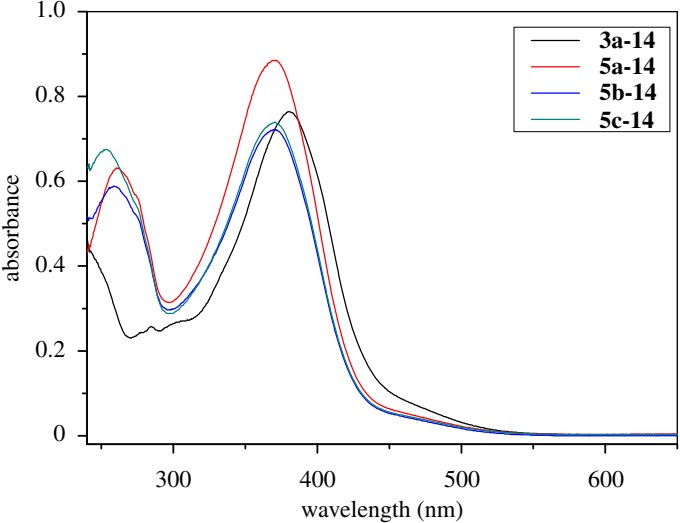

**Figure 3.** UV–Vis spectra of selected compounds in CH$_2$Cl$_2$ (10$^{-5}$ M).

during the cooling process, compound **5c-14** underwent a transition from isotropic liquid to mesomorphic phase at 85°C. The mesomorphic phase lasted up to 65°C, and the mesomorphic phase interval was 20°C. When the cooling continued, the compound solidified and entered the crystalline phase. The comparison images of **5c-14** at 81.0°C in mesomorphic phase and 44.7°C in crystalline phase on cooling are shown in figure 2g and h. So lengthening the terminal alkyl chain on both sides of the molecular can also induce liquid crystallinity.

## 2.2. Photoisomerization studies

The photoisomerization of selective compounds was studied. At first, the UV–visible absorption spectra (UV–Vis) of unexposed materials were measured in CH$_2$Cl$_2$, and the obtained results are shown in figure 3. Two absorption bands were observed for all selected compounds. The absorption band at 380 nm for compound **3a-14** was assigned to the $\pi$–$\pi^*$ electronic transitions of *trans* isomers in azo compounds [37], which shifted hypsochromicly about 10 nm for compounds **5a–5c**. This is due to the fact that the alkoxy group connected with azobenzene in **3a** series compounds have stronger electron-donating ability than the benzoic ester groups connected with azobenzene in **5a–5c** and is better auxochrome.

Next, the selected samples **3a-14** and **5b-14** were irradiated using wavelength of 365 nm for a certain period of time, and then their UV–Vis absorption spectrum changes were observed. As shown in figure 4a and c, the absorption band at 370–380 nm gradually decreased with the increase of irradiation time. The change of absorption spectra is due to the decrease of azobenzene E isomers in the sample. Meanwhile, new absorption bands at 440 nm (*n*-$\pi^*$ electron transition) and 300–330 nm for compounds **3a-14** and **5b-14**, and 260 nm for compound **3a-14** appeared and the absorption intensity increased gradually, which indicated the increase of azobenzene Z isomers in the sample [8]. At the same time, three isosbestic points at 257, 340 and 448 nm for **3a-14** and 253, 325 and 436 nm for **5b-14** were observed, which showed that there were no side reactions such as photodegradation except photoisomerization [38]. Table 2 lists the time taken for **3a-14** and **5b-14** for light isomerization and thermal back relaxation to reach equilibrium with conversion efficiency (CE) of E–Z photoisomerization. The CE value is estimated from the following equation [8]:

$$\text{CE} = \frac{A_0 - A_t}{A_0} \times 100\%,$$

where $A_0$ and $A_t$ are absorbance before UV and after UV, respectively. As shown in figure 4 and table 2, the conversion between E and Z isomers in azobenzene reached equilibrium after UV irradiation. **3a-14** took 25 s to reach equilibrium of E–Z transition with CE value of 74.4%, whereas **5b-14** took longer time (32 s) to reach equilibrium with the low CE of 57.3%.

After irradiation, thermal back relaxation of samples was investigated in the dark. As shown in figure 4b and d, in contrast with exposure under UV light, the absorption peaks of $\pi$–$\pi^*$ transition near 370–380 nm increased gradually and the absorption peaks at 260, 300–330 and 480 nm decreased

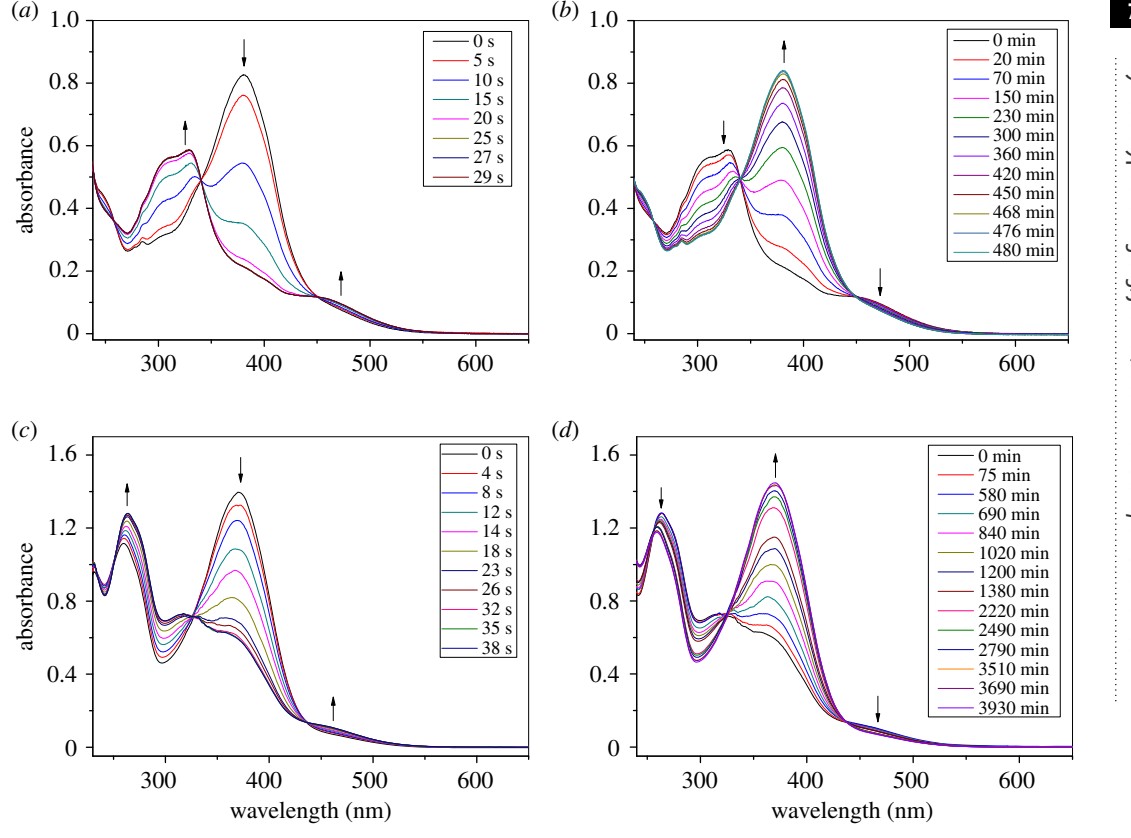

**Figure 4.** Changes of absorption spectra of **3a-14** (*a,b*) and **5b-14** (*c,d*) in CH₂Cl₂ during UV exposure (*a,c*) and thermal back relaxation (*b,d*).

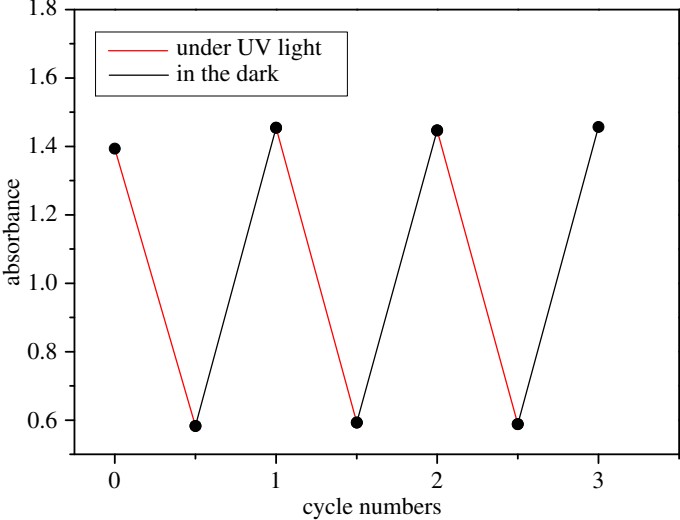

**Figure 5.** Three cycles of UV–Vis absorbance at 370 nm for compound **5b-14**.

**Table 2.** Time taken for **3a-14** and **5b-14** during UV light isomerization and thermal back relaxation to reach equilibrium with conversion efficiency.

| compd. | E–Z (Time) (s) | Z–E (Time) (h) | CE (%) |
|---|---|---|---|
| **3a-14** | 25 | 8 | 74.4 |
| **5b-14** | 32 | 61.5 | 57.3 |

gradually indicating Z isomer converts back to E form for **3a-14** and **5b-14**. Remarkably, thermal back relaxation was much slower than the isomerization exposed under UV light. More specifically, it took 8 h for **3a-14** and 61.5 h for **5b-14** to recover to the original state. Long isomerization time and low CE of the latter indicate that compound **5b-14** might tend to exist in E isomers, which may be due to the larger steric hindrance of the alkoxybenzoate linked to azobenzene in compound **5b-14** compared with the alkoxy group in **3a-14**.

The photoisomerization behaviours of the azobenzene derivatives can be recycled many times. Figure 5 shows the minimum and maximum absorbance changes of **5b-14** at 370 nm after three cycles of exposure and thermal back relaxation in the dark. After each UV irradiation, the absorption can be recovered to its initial state in the dark. We expect that these compounds also display the photoisomerization properties in solid state and plan to further study their potential application in the field of optical switch and optical information storage [39].

## 3. Conclusion

A series of new azobenzene compounds with dihydropyrazole heterocycle were synthesized and characterized. We found that the compounds containing four linearly linked rings have no liquid crystallinity. In the case of the azobenzene derivatives containing five linearly linked rings, lengthening the terminal alkyl chain near ester group or on both sides of the molecular can induce liquid crystallinity of the molecules. Selected azo compounds underwent isomerization from E to Z under ultraviolet irradiation and then thermal back relaxation slowly in the dark. The photoisomerization behaviours of the azobenzene derivatives can be recycled many times, which is worthy of further study in the field of optical switch and optical information storage.

## 4. Experimental

### 4.1. Material and measurements

Compounds **1a–1c** and acyl chloride **4** were synthesized according to the literature [35]. **2a–2c** were new compounds and their synthesis and characterization are described in electronic supplementary material. UltrafleXtreme mass spectrometer (MALDI-TOF/TOF) and ALPHA spectrometer of Bruker were used to measure HRMS and FTIR (KBr pellets) spectra, respectively. Avance 500 Bruker and Shimadzu UV2600 spectrometer were used to measure NMR spectra and Electronic absorption spectra, respectively. METTLER TOLEDO DSC3 was used to measure DSC thermographs under $N_2$ environment at a cooling and heating rate of 5°C min$^{-1}$. POM images were observed on a Leica DM4500p with a LTS420 Freezing and Heating stage system.

### 4.2. General procedure for the synthesis of compounds 3 and 5

Compound **2** (0.3 mmol) was dissolved in dry pyridine (2 ml), to which the pyridine solution (2 ml) of bromoalkane (0.5 mmol) or acyl chloride **4** was slowly added at 5°C. After stirring 4 h at room temperature, pyridine was removed by vacuum distillation. The residue was redissolved in $CH_2Cl_2$, washed with water, dried over anhydrous $Na_2SO_4$ and distilled to remove the solution. The residue was purified by column chromatography over silica gel eluted with petroleum ether/EtOAc (5 : 1, V: V). The second fraction was collected, and then recrystallized from $CH_2Cl_2$/n-hexane to afford the pure compounds **3** and **5**.

**3a-8**. Yield 71%, m.p. 97–99°C; $^1$H NMR (500 MHz, CDCl$_3$) $\delta$: 7.95 (t, $J$ = 8.5 Hz, 4H), 7.88 (d, $J$ = 8.5 Hz, 2H), 7.19 (d, $J$ = 9.0 Hz, 2H), 7.02 (d, $J$ = 9.0 Hz, 2H), 6.85 (d, $J$ = 9.0 Hz, 2H), 5.59 (dd, 1H), 4.05 (t, $J$ = 6.5 Hz, 2H), 3.77 (s, 3H), 3.20 (dd, 1H), 2.44 (s, 3H), 2.32 (s, 3H), 1.84–1.81 (m, 2H), 1.48–1.45 (m, 2H), 1.39–1.29 (m, 8H), 0.89 (t, $J$ = 6.5 Hz, 3H) ppm; $^{13}$C NMR (125 MHz, CDCl$_3$) $\delta$: 168.99, 162.24, 159.17, 153.65, 153.22, 146.99, 134.13, 133.12, 127.45, 127.06, 125.14, 123.05, 114.90, 114.37, 68.54, 59.78, 55.39, 42.36, 31.93, 29.47, 29.36, 29.30, 26.14, 22.78, 22.17, 14.24 ppm; IR (KBr) $v$: 2919, 2852, 1679, 1598, 1398, 1253, 1139, 1024, 844 cm$^{-1}$; HRMS $m/z$: Calcd for C$_{32}$H$_{38}$N$_4$O$_3$Na 549.2836 [M + Na]$^+$, found 549.2843.

**3a-14**. Yield 70.6%, m.p. 97–98°C; $^1$H NMR (500 MHz, CDCl$_3$) $\delta$: 7.95 (d, $J$ = 8.5 Hz, 2H), 7.93 (d, $J$ = 8.5 Hz, 2H), 7.87 (d, $J$ = 8.5 Hz, 2H), 7.18 (d, $J$ = 8.5 Hz, 2H), 7.02 (d, $J$ = 8.5 Hz, 2H), 7.86 (d, $J$ = 8.5 Hz, 2H), 5.59 (dd, 1H), 4.05 (t, $J$ = 6.5 Hz, 2H), 3.77 (s, 3H), 3.20 (dd, 1H), 2.44 (s, 3H), 1.84–1.81 (m, 2H),

1.48–1.46 (m, 2H), 1.37–1.26 (m, 20H), 0.89 (t, $J$ = 7.0 Hz, 3H) ppm; $^{13}$C NMR (125 MHz, CDCl$_3$) $\delta$: 169.01, 162.25, 159.18, 153.66, 153.22, 147.00, 134.14, 133.13, 127.46, 127.07, 125.15, 123.06, 114.91, 114.38, 68.55, 59.78, 55.40, 42.37, 32.06, 29.79, 29.73, 29.50, 29.30, 26.13, 22.83, 22.18, 14.26 ppm; IR (KBr) $v$: 2921, 2848, 1677, 1581, 1405, 1247, 1143, 1016, 848 cm$^{-1}$; HRMS $m/z$: Calcd for C$_{38}$H$_{51}$N$_4$O$_3$ 611.3956 [M + H]$^+$, found 611.3961.

**5a-8**. Yield 55%, m.p. 139–142°C; $^1$H NMR (CDCl$_3$, 500 MHz) $\delta$: 8.16 (d, $J$ = 9.0 Hz, 2H), 8.03 (d, $J$ = 8.5 Hz, 2H), 7.97 (d, $J$ = 8.5 Hz, 2H), 7.89 (d, $J$ = 8.5 Hz, 2H), 7.39 (d, $J$ = 8.5 Hz, 2H), 7.19 (d, $J$ = 9.0 Hz, 2H), 6.99 (d, $J$ = 9.0 Hz, 2H), 6.86 (d, $J$ = 8.5 Hz, 2H), 5.58 (dd, 1H), 4.05 (t, $J$ = 6.5 Hz, 2H), 3.78 (m, 4H), 3.23 (dd, 1H), 2.44 (s, 3H), 1.86–1.80 (m, 2H), 1.51–1.45 (m, 2H), 1.39–1.26 (m, 8H), 0.89 (t, $J$ = 6.75 Hz, 3H) ppm; $^{13}$C NMR (CDCl$_3$, 125 MHz) $\delta$: 169.00, 162.25, 159.18, 153.67, 153.22, 147.01, 134.14, 133.13, 127.45, 127.07, 125.15, 123.06, 114.91, 114.38, 68.55, 59.79, 55.40, 42.37, 31.94, 29.47, 29.36, 29.30, 26.14, 22.79, 22.17, 14.24 ppm; IR (KBr) $v$: 3054, 2923, 2856, 1722, 1664, 1602, 1511, 1409, 1257, 1159, 1074, 1024, 842 cm$^{-1}$; HRMS $m/z$: Calcd for C$_{39}$H$_{43}$N$_4$O$_5$ [M + H]$^+$ 647.3228, found 647.3200.

**5a-10**. Yield 55%, m.p. 123–125°C; $^1$H NMR (CDCl$_3$, 500 MHz) $\delta$: 8.16 (d, $J$ = 9.0 Hz, 2H), 8.03(d, $J$ = 8.5 Hz, 2H), 7.97(d, $J$ = 8.5 Hz, 2H), 7.89 (d, $J$ = 8.5 Hz,2H), 7.39 (d, $J$ = 8.5 Hz, 2H), 7.19(d, $J$ = 9.0 Hz, 2H), 6.99 (d, $J$ = 9.0 Hz, 2 H), 6.86 (d, $J$ = 8.5 Hz, 2H), 5.52 (dd, 1H), 4.05 (t, $J$ = 6.5 Hz, 2H), 3.81–3.75 (m, 4H), 3.23 (dd, 1H), 2.44 (s, 3H), 1.85–1.80 (m, 2H), 1.49–1.45 (m, 2H), 1.35–1.26 (m, 12H), 0.89 (t, $J$ = 6.75 Hz, 3H) ppm; $^{13}$C NMR (CDCl$_3$, 125 MHz) $\delta$: 169.08, 164.73, 163.90, 159.22, 153.66, 153.40, 153.10, 150.31, 134.11, 133.90, 132.53, 127.52, 127.09, 124.44, 123.42, 122.71, 121.26, 114.54, 114.42, 68.52, 59.87, 55.43, 42.37, 32.04, 29.70, 29.50, 29.46, 29.23, 26.12, 22.83, 22.19, 14.27 ppm; IR (KBr) $v$: 3056, 2921, 2854, 1722, 1666, 1604, 1511, 1257, 1164, 846 cm$^{-1}$; HRMS $m/z$: Calcd for C$_{41}$H$_{46}$N$_4$O$_5$Na 697.3360 [M + Na]$^+$, found 697.3378.

**5a-12**. Yield 53.1%, m.p. 125–128°C; $^1$H NMR (CDCl$_3$, 500 MHz) $\delta$: 8.16 (d, $J$ = 9.0 Hz, 2H), 8.03 (d, $J$ = 8.5 Hz, 2H), 7.97 (d, $J$ = 8.5 Hz, 2H), 7.89 (d, $J$ = 8.5 Hz,2H), 7.39 (d, $J$ = 8.5 Hz, 2H), 7.19 (d, $J$ = 9.0 Hz, 2H), 6.99 (d, $J$ = 9.0 Hz, 2H), 6.86 (d, $J$ = 8.5 Hz, 2H), 5.58 (dd, 1H), 4.05 (t, $J$ = 6.5 Hz, 2H), 3.86–3.74 (t, 4H), 3.22 (dd, 1H), 2.44 (s, 3H), 1.85–1.80 (m, 2H), 1.50–1.45 (m, 2H), 1.38–1.24 (m, 16H), 0.89 (t, $J$ = 7.0 Hz, 3H) ppm; $^{13}$C NMR (CDCl$_3$, 125 MHz) $\delta$: 169.04, 164.70, 163.88, 159.20, 153.64, 153.37, 153.06, 150.28, 134.10, 133.89, 132.51, 127.50, 127.07, 124.42, 123.40, 122.69, 121.24, 114.52, 114.40, 68.50, 59.85, 55.40, 42.35, 32.05, 29.79, 29.77, 29.72, 29.69, 29.48, 29.21, 26.11, 22.82, 22.18, 14.26 ppm; IR (KBr) $v$: 2917, 2848, 1725, 1679, 1608, 1513, 1400, 1265, 1170, 1072, 848 cm$^{-1}$; HRMS $m/z$: Calcd for C$_{43}$H$_{51}$N$_4$O$_5$ 703.3854 [M + H]$^+$, found 703.3842.

**5a-14**. Yield 53.6%, m.p. 123–125°C; $^1$H NMR (CDCl$_3$, 500 MHz) $\delta$: 8.16 (d, $J$ = 9.0 Hz, 2H), 8.03 (d, $J$ = 8.5 Hz, 2H), 7.97 (d, $J$ = 8.5 Hz, 2H), 7.89 (d, $J$ = 8.5 Hz, 2H), 7.39 (d, $J$ = 8.5 Hz, 2H), 7.19 (d, $J$ = 9.0 Hz, 2H), 6.99 (d, $J$ = 9.0 Hz, 2H), 6.86 (d, $J$ = 8.5 Hz, 2H), 5.58 (dd, 1H), 4.05 (t, $J$ = 6.5 Hz, 2H), 3.81–3.75 (m, 4H), 3.23 (dd, 1H), 2.44 (s, 3H), 1.85–1.80 (m, 2H), 1.49–1.45 (m, 2H), 1.39–1.26 (m, 20H), 0.88 (t, $J$ = 7.0 Hz, 3H) ppm; $^{13}$C NMR (CDCl$_3$, 125 MHz) $\delta$: 169.05, 164.71, 163.89, 159.21, 153.65, 153.39, 153.07, 150.30, 134.11, 133.89, 132.52, 127.51, 127.08, 124.43, 123.41, 122.69, 121.25, 114.53, 114.41, 68.51, 59.86, 55.41, 42.36, 32.06, 29.83, 29.81, 29.79, 29.73, 29.70, 29.50, 29.22, 26.12, 22.83, 22.19, 14.27 ppm; IR (KBr) $v$: 3050, 2917, 2848, 1729, 1670, 1608, 1513, 1396, 1263, 1170, 1076, 848 cm$^{-1}$; HRMS $m/z$: Calcd for C$_{45}$H$_{54}$N$_4$O$_5$Na 753.3986 [M + Na]$^+$, found 753.3934.

**5a-16**. Yield 53.1%, m.p. 121–123°C; $^1$H NMR (CDCl$_3$, 500 MHz) $\delta$: 8.16 (d, $J$ = 9.0 Hz, 2H), 8.03 (d, $J$ = 8.5 Hz, 2H), 7.97 (d, $J$ = 8.5 Hz, 2H), 7.89 (d, $J$ = 8.5 Hz, 2H), 7.39 (d, $J$ = 8.5 Hz, 2H), 7.19 (d, $J$ = 9.0 Hz, 2H), 6.99 (d, $J$ = 9.0 Hz, 2H), 6.86 (d, $J$ = 8.5 Hz, 2H), 5.58 (dd, 1H), 4.05 (t, 4H), 3.22 (dd, 1H), 2.44 (s, 3H), 1.85–1.79 (m, 2H), 1.50–1.45 (m, 2H), 1.38–1.26 (m, 24H), 0.89 (t, $J$ = 7.0 Hz, 3H) ppm; $^{13}$C NMR (CDCl$_3$, 125 MHz) $\delta$: 169.05, 164.70, 163.88, 159.20, 153.64, 153.37, 153.08, 150.28, 134.09, 133.88, 132.51, 127.50, 127.07, 124.42, 123.40, 122.68, 121.24, 114.52, 114.40, 68.50, 59.85, 55.40, 42.35, 32.06, 29.83, 29.79, 29.72, 29.69, 29.50, 29.22, 26.11, 22.82, 22.18, 14.26 ppm; IR (KBr) $v$: 2919, 2848, 1729, 1675, 1604, 1509, 1257, 1164, 1070, 842 cm$^{-1}$; HRMS $m/z$: Calcd for C$_{47}$H$_{59}$N$_4$O$_5$ 759.4480 [M + H]$^+$, found 759.4489.

**5b-10**. Yield 43%, m.p. 128–130°C; $^1$H NMR (CDCl$_3$, 500 MHz) $\delta$: 8.16 (d, $J$ = 9.0 Hz, 2H), 8.03 (d, $J$ = 8.5 Hz, 2H), 7.97 (d, $J$ = 8.5 Hz, 2H), 7.89 (d, $J$ = 8.5 Hz, 2H), 7.39 (d, $J$ = 8.5 Hz, 2H), 7.16 (d, $J$ = 9.0 Hz, 2H), 6.99 (d, $J$ = 9.0 Hz, 2H), 6.86 (d, $J$ = 8.5 Hz, 2H), 5.58 (dd, 1H), 4.05 (t, $J$ = 6.5 Hz, 2H), 3.89 (t, $J$ = 6.5 Hz, 2H), 3.77 (dd, 1H), 3.20 (dd, 1H), 2.43 (s, 3H), 1.88–1.80 (m, 4H), 1.77–1.72 (m, 2H), 1.51–1.41 (m, 4H), 1.37–1.26 (m, 12H), 0.90–0.87 (m, 6H) ppm; $^{13}$C NMR (CDCl$_3$, 125 MHz) $\delta$: 169.08, 164.71, 163.90, 158.83, 153.65, 153.39, 153.12, 150.31, 133.92, 133.84, 132.52, 127.51, 127.03, 124.42, 123.41, 122.69, 121.27, 114.96, 114.53, 68.52, 68.15, 59.89, 42.37, 32.03, 31.94, 29.68, 29.50, 29.47, 29.45, 29.36, 29.22, 26.17, 26.11, 22.81, 22.78, 22.17, 14.25, 14.23 ppm; IR (KBr) $v$: 3054, 2921, 2854, 1724, 1670, 1602, 1419, 1257, 1076, 802 cm$^{-1}$; HRMS $m/z$: Calcd for C$_{44}$H$_{53}$N$_4$O$_5$ 717.4010 [M + H]$^+$, found 717.3953.

**5b-14**. Yield 42.6%, m.p. 102–104°C; $^1$H NMR (CDCl$_3$, 500 MHz) $\delta$: 8.16 (d, $J$ = 9.0 Hz, 2H), 8.03 (d, $J$ = 8.5 Hz, 2H), 7.97 (d, $J$ = 8.5 Hz, 2H), 7.89 (d, $J$ = 8.5 Hz,2H), 7.39 (d, $J$ = 8.5 Hz, 2H), 7.19 (d, $J$ = 9.0 Hz, 2H), 6.99 (d, $J$ = 9.0 Hz, 2H), 6.84 (d, $J$ = 8.5 Hz, 2H), 5.57 (dd, 1H), 4.05 (t, $J$ = 6.5 Hz, 2H), 3.92 (t, $J$ = 6.5 Hz, 2H), 3.76 (dd, 1H), 3.22 (dd, 1H), 2.44 (s, 3H), 1.85–1.78 (m, 2H), 1.76–1.71 (m, 2H), 1.49–1.43 (m, 4H), 1.36–1.26 (m, 20H), 0.95 (t, $J$ = 7.5 Hz, 3H), 0.88 (t, $J$ = 7.0 Hz, 3H) ppm; $^{13}$C NMR (CDCl$_3$, 125 MHz) $\delta$: 169.10, 164.71, 163.90, 158.83, 153.66, 153.39, 153.13, 150.31, 133.92, 133.84, 132.52, 127.51, 127.03, 124.43, 123.41, 122.69, 121.27, 114.96, 114.53, 114.30, 68.52, 67.82, 59.89, 42.37, 32.06, 31.42, 29.83, 29.81, 29.79, 29.73, 29.70, 29.50, 29.23, 26.12, 22.83, 22.18, 19.36, 14.26, 13.96 ppm; IR (KBr) $v$: 3060, 2919, 2850, 1729, 1677, 1608, 1465, 1263, 1168, 1072, 848 cm$^{-1}$; HRMS $m/z$: Calcd for C$_{48}$H$_{61}$N$_4$O$_5$ 773.4636 [M + H]$^+$, found 773.4530.

**5c-10**. Yield 42.7%, m.p. 133–135°C; $^1$H NMR (CDCl$_3$, 500 MHz) $\delta$: 8.16 (d, $J$ = 9.0 Hz, 2H), 8.03 (d, $J$ = 8.5 Hz, 2H), 7.97 (d, $J$ = 8.5 Hz, 2H), 7.89 (d, $J$ = 8.5 Hz, 2H), 7.39 (d, $J$ = 8.5 Hz, 2H), 7.17 (d, $J$ = 9.0 Hz, 2 H), 6.99 (d, $J$ = 9.0 Hz, 2H), 6.84 (d, $J$ = 8.5 Hz, 2H), 5.57 (dd, 1H), 4.04 (t, $J$ = 6.5 Hz, 2H), 3.92 (t, $J$ = 6.5 Hz, 2H), 3.77 (dd 1H), 3.22 (dd, 1H), 2.43 (s, 3H), 1.85–1.80 (m, 2H), 1.75–1.71 (m, 2H), 1.49–1.46 (m, 4H), 1.32–1.28 (m, 20H), 0.95 (t, $J$ = 7.5 Hz, 3H), 0.88 (t, $J$ = 7.0 Hz, 3H) ppm; $^{13}$C NMR (CDCl$_3$, 125 MHz) $\delta$: 169.07, 164.71, 163.89, 158.82, 153.65, 153.39, 153.11, 150.30, 133.92, 133.84, 132.51, 127.51, 127.03, 124.42, 123.40, 122.69, 121.26, 114.96, 114.53, 68.51, 67.81, 59.89, 53.56, 42.36, 32.03, 31.42, 29.68, 29.49, 29.45, 29.22, 26.11, 22.81, 22.17, 19.36, 14.25, 13.95 ppm; IR (KBr) $v$: 3048, 2923, 2854, 1729, 1646, 1604, 1506, 1257, 1166, 1066, 842 cm$^{-1}$; HRMS $m/z$: Calcd for C$_{48}$H$_{61}$N$_4$O$_5$ 773.4642 [M + H]$^+$, found 773.4522.

**5c-14**. Yield 55%, m.p. 94–96°C; $^1$H NMR (CDCl$_3$, 500 MHz) $\delta$: 8.16 (d, $J$ = 9.0 Hz, 2H), 8.03 (d, $J$ = 8.5 Hz, 2H), 7.97 (d, $J$ = 8.5 Hz, 2H), 7.89 (d, $J$ = 8.5 Hz, 2H), 7.39 (d, $J$ = 8.5 Hz, 2H), 7.17 (d, $J$ = 9.0 Hz, 2H), 6.99 (d, $J$ = 9.0 Hz, 2H), 6.86 (d, $J$ = 8.5 Hz, 2H), 5.59 (dd, 1H), 4.05 (t, $J$ = 6.5 Hz, 2H), 3.89 (t, $J$ = 6.5 Hz, 2H), 3.77 (dd, 1H), 3.20 (dd, 1H), 2.43 (s, 3H), 1.84–1.73 (m, 4H), 1.49–1.27 (m, 32H), 0.89 (m, 6H) ppm; $^{13}$C NMR (CDCl$_3$, 125 MHz) $\delta$: 171.31, 169.17, 164.72, 163.90, 163.75, 158.83, 153.66, 153.40, 153.22, 150.30, 133.90, 133.81, 132.52, 132.42, 127.52, 127.03, 124.42, 123.41, 122.69, 121.56, 121.26, 114.96, 114.53, 114.30, 68.52, 68.41, 68.15, 59.90, 42.38, 32.06, 31.94, 29.83, 29.81, 29.79, 29.73, 29.69, 29.50, 29.47, 29.37, 29.23, 26.17, 26.12, 22.83, 22.78, 22.16, 14.26, 14.23 ppm; IR (KBr) $v$: 2919, 2850, 1727, 1675, 1606, 1427, 1519, 1259, 1168, 1072, 846 cm$^{-1}$; HRMS $m/z$: Calcd for C$_{52}$H$_{69}$N$_4$O$_5$ 829.5268 [M + H]$^+$, found 829.5090.

Data accessibility. All data used in this article are present in the article and its electronic supplementary material.

Authors' contributions. X.W. and Z.L. carried out the laboratory work and participated in data analysis. H.Z. conceived of and designed the experiments and drafted the manuscript. S.C. collected the field data and helped to draft the manuscript. All authors gave final approval for publication.

Competing interests. We declare we have no competing interests.

Funding. This work was supported by the National Natural Science Foundation of China (grant nos. 21961023 and 21562032), Natural Science Foundation of Inner Mongolia (grant no. 2017MS0205).

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
