## [Reviewer comments · Royal Society Open Science]

Review History

RSOS-200474.R0 (Original submission)

Review form: Reviewer 1

Is the manuscript scientifically sound in its present form?

Yes

Are the interpretations and conclusions justified by the results?

Yes

Is the language acceptable?

Yes

Do you have any ethical concerns with this paper?

No

Have you any concerns about statistical analyses in this paper?

Yes

Recommendation?

Accept with minor revision (please list in comments)

Comments to the Author(s)

In this manuscript, the authors prepared and characterized a series of new azobenzene compounds with dihydropyrazole heterocycle. The authors found the azobenzene derivatives containing five linearly linked rings, lengthening the terminal alkyl chain near ester group or on both sides of the molecular can induce liquid crystallinity of the molecules. This research is carefully conducted and all data are reasonable. I would like to recommend its publication after minor revisions.

- (1) The original HRMS should be provided in SI.
- (2) Any CV studied for the as-prepared compounds?
- (3) Some heterocyclic references might be included in the revised manuscript: Chemical Record, 2016, 16, 1518; Chem. Mater. 2017, 29, 4172; Mater Chem Front, 2020, DOI: 10.1039/C9QM00656G

Review form: Reviewer 2

Is the manuscript scientifically sound in its present form?

Yes

Are the interpretations and conclusions justified by the results?

Yes

Is the language acceptable?

Yes

Do you have any ethical concerns with this paper?

No

Have you any concerns about statistical analyses in this paper?

No

Recommendation?

Accept with minor revision (please list in comments)

Comments to the Author(s)

In the manuscript, the authors reported new azobenzene derivatives with dihydropyrazole heterocycle, and systematically investigated their liquid crystal properties and photoisomerisation properties. The manuscript is generally well written and logically organized, though it lacks scientific insights a little bit. It could be considered for publication after revisions as follows:

1. It would be better if the authors could define the liquid crystalline phases in Fig. 2.
2. Other reports regarding photoisomerisation of azobenzene derivatives should be provided, for example, J. Am. Chem. Soc. 2020, 142, 6467-6471.

Decision letter (RSOS-200474.R0)

Dear Dr Zhao:

Title: New azobenzene liquid crystal with dihydropyrazole heterocycle and photoisomerisation studies

Manuscript ID: RSOS-200474

Thank you for submitting the above manuscript to Royal Society Open Science. On behalf of the Editors and the Royal Society of Chemistry, I am pleased to inform you that your manuscript will be accepted for publication in Royal Society Open Science subject to minor revision in accordance with the referee suggestions. Please find the reviewers' comments at the end of this email.

The reviewers and handling editors have recommended publication, but also suggest some minor revisions to your manuscript. Therefore, I invite you to respond to the comments and revise your manuscript.

Because the schedule for publication is very tight, it is a condition of publication that you submit the revised version of your manuscript before 22-May-2020. Please note that the revision deadline will expire at 00.00am on this date. If you do not think you will be able to meet this date please let me know immediately.

Kind regards,
Dr Laura Smith
Publishing Editor, Journals

On behalf of the Subject Editor Professor Anthony Stace and the Associate Editor Dr Andrew Harned.

RSC Associate Editor:

Comments to the Author:

The referee reports are short, but consider the scientific aspects of the work to be suitable for publication in RSOS. They have suggested a few minor revisions that should be addressed before final acceptance.

RSC Subject Editor:

Comments to the Author:

(There are no comments.)

Reviewer comments to Author:

Reviewer: 1

Comments to the Author(s)

In this manuscript, the authors prepared and characterized a series of new azobenzene compounds with dihydropyrazole heterocycle. The authors found the azobenzene derivatives containing five linearly linked rings, lengthening the terminal alkyl chain near ester group or on both sides of the molecular can induce liquid crystallinity of the molecules. This research is carefully conducted and all data are reasonable. I would like to recommend its publication after minor revisions.

(1) The original HRMS should be provided in SI.

(2) Any CV studied for the as-prepared compounds?

(3) Some heterocyclic references might be included in the revised manuscript: Chemical Record, 2016, 16, 1518; Chem. Mater. 2017, 29, 4172; Mater Chem Front, 2020, DOI: 10.1039/C9QM00656G

Reviewer: 2

Comments to the Author(s)

In the manuscript, the authors reported new azobenzene derivatives with dihydropyrazole heterocycle, and systematically investigated their liquid crystal properties and photoisomerisation properties. The manuscript is generally well written and logically organized, though it lacks scientific insights a little bit. It could be considered for publication after revisions as follows:

1. It would be better if the authors could define the liquid crystalline phases in Fig. 2.
2. Other reports regarding photoisomerisation of azobenzene derivatives should be provided, for example, J. Am. Chem. Soc. 2020, 142, 6467-6471.

Author's Response to Decision Letter for (RSOS-200474.R0)

See Appendix A.

Decision letter (RSOS-200474.R1)

Dear Dr Zhao:

Title: New azobenzene liquid crystal with dihydropyrazole heterocycle and photoisomerisation studies

Manuscript ID: RSOS-200474.R1

It is a pleasure to accept your manuscript in its current form for publication in Royal Society Open Science. The chemistry content of Royal Society Open Science is published in collaboration with the Royal Society of Chemistry.

On behalf of the Subject Editor Professor Anthony Stace and the Associate Editor Dr Andrew Harned.

RSC Associate Editor
Comments to the Author:

The authors have addressed the reviewer's comments as best as they could given the focus of this manuscript. I believe this work is now suitable for publication.

Reviewer(s)' Comments to Author:

Appendix A

Response to the comments

<Journal Name> **Royal Society Open Science**

<Manuscript Title> **New azobenzene liquid crystal with dihydropyrazole heterocycle and photoisomerisation studies**

Manuscript ID: **RSOS-200474**

Thank you for your useful comments and suggestions. We have modified the manuscript accordingly, and detailed corrections are listed below point by point:

Reviewer #1:

In this manuscript, the authors prepared and characterized a series of new azobenzene compounds with dihydropyrazole heterocycle. The authors found the azobenzene derivatives containing five linearly linked rings, lengthening the terminal alkyl chain near ester group or on both sides of the molecular can induce liquid crystallinity of the molecules. This research is carefully conducted and all data are reasonable. I would like to recommend its publication after minor revisions.

(1) The original HRMS should be provided in SI.

Response: The original HRMS have been provided in SI (Fig. S35-S45)

(2) Any CV studied for the as-prepared compounds?

Response : Thank you very much for your advice. In this manuscript, azobenzene compounds with dihydropyrazole heterocycle were prepared, and liquid crystal and photoisomerisation of these compounds were studied. The electrochemical properties (CV) have not been studied at present, and we are going to study the CV in the follow-up work.

(3) Some heterocyclic references might be included in the revised manuscript: Chemical Record, 2016, 16, 1518; Chem. Mater. 2017, 29, 4172; Mater Chem Front, 2020, DOI: 10.1039/C9QM00656G

Response: The above references have been cited (see references [21] - [23]).

Reviewer #2:

In the manuscript, the authors reported new azobenzene derivatives with dihydropyrazole heterocycle, and systematically investigated their liquid crystal properties and photoisomerisation properties. The manuscript is generally well written and logically organized, though it lacks scientific insights a little bit. It could be considered for publication after revisions as follows:

1. It would be better if the authors could define the liquid crystalline phases in Fig. 2.

Response: It's a pity that it is difficult to identify the mesophases with the existing experimental data.

2. Other reports regarding photoisomerisation of azobenzene derivatives should be provided, for example, J. Am. Chem. Soc. 2020, 142, 6467-6471.

Response: The above reference has been cited (see reference [41]).